# Thermal Performance of Double-Pane Lightweight Steel Framed Walls with and without a Reflective Foil

**Paulo Santos ***[ID] **and Telmo Ribeiro** [ID]

ISISE, Department of Civil Engineering, University of Coimbra, 3030-788 Coimbra, Portugal;
telmo.ribeiro@dec.uc.pt
* Correspondence: pfsantos@dec.uc.pt

**Abstract:** One strategy to increase energy efficiency of buildings could be the reduction of undesirable heat losses by mitigating the heat transfer mechanisms across the building envelope. The use of thermal insulation is the simplest and most straightforward way to promote thermal resistance of building elements by reducing the heat transfer by conduction. However, whenever there is an air cavity, radiation heat transfer could be also very relevant. The use of thermal reflective insulation materials inside the air gaps of building elements is likewise an effective way to increase thermal resistance without increasing weight and wall thickness. Some additional advantages are its low-cost and easy installation. In this work, the performance of a thermal reflective insulation system, constituted by an aluminium foil placed inside an air cavity between a double pane lightweight steel framed (LSF) partition, is experimentally evaluated for different air gap thicknesses, ranging from 0 mm up to 50 mm, with a step increment of 10 mm. We found a maximum thermal resistance improvement of the double pane LSF walls due to the reflective foil of around +0.529 $m^2 \cdot °C/W$ (+21%). The measurements of the *R*-values were compared with predictions provided by simplified models (CEN and NFRC 100). Both models were able to predict with reasonable accuracy (around ±5%) the thermal behaviour of the air cavities within the evaluated double pane LSF walls.

**Keywords:** thermal performance; experimental assessment; simplified models; double-pane; lightweight steel frame (LSF); partition walls; aluminium reflective foil

---

## 1. Introduction

Buildings are a major key sector regarding energy consumption. In the European Union (EU), almost 50% of final energy consumption is used for heating and cooling, of which 80% is used in buildings [1]. Moreover, the building stock is responsible for approximately 36% of all $CO_2$ emissions in the EU. Consequently, in order to achieve highly energy efficient and decarbonised buildings, European countries have established ambitious climate and energy targets, guaranteeing the conversion of existing building stock into nearly zero-energy buildings (NZEB's) [1], and developing the use of renewable energy sources (e.g., solar) [2], thus implementing long-term refurbishment strategies.

One possible strategy to improve energy efficiency of buildings is the reduction of undesirable heat losses, by mitigating each heat transfer mechanism across the building envelope [3]: conduction, convection and radiation. The simplest and most straightforward way to promote thermal resistance of building elements is the use of thermal insulation, which significantly reduces heat transfer by conduction, although its effectiveness depends also on their position within the building element [4]. Moreover, this thermal insulation material also promotes sound insulation, mainly when fibrous insulation materials are used inside the air gaps [5].

Nowadays, highly efficient insulation materials are emerging with very low thermal conductivities, which are designated as super insulating materials (SIMs) [6]. Two common examples are vacuum insulating panels (VIPs) [7] and aerogels [8]. Nevertheless, the inadequate building design and use of these SIMs may contribute to increasing thermal

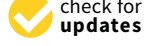

bridges' importance. In fact, thermal bridges are very relevant for thermal behaviour and energy efficiency of buildings, and could be responsible for up to almost one third of the heating energy needs [9]. As such, when using structural solutions containing materials with high thermal conductivity, such as steel, the significance of thermal bridges could be even greater [10].

Over recent years, the lightweight steel frame (LSF) construction system has attracted attention from buildings' stockholders, mainly for low-rise residential houses, due to their inherent advantages [11]. Among these benefits are low weight, high mechanical strength, fast construction, reduced on-site disruption, high potential for recycling and reuse, high architectural flexibility, suitability for retrofitting, easy prefabrication, economical transportation and handling, superior quality, precise tolerances, high quality standards, humidity stability shape, and insect damage resistance.

Currently, there are several strategies to mitigate thermal bridges in LSF buildings' components, such as thermal break strips [12–14], slotted steel studs [12,15,16] and continuous external thermal insulation composite system (ETICS) [17,18]. Notice that even slight modifications in the steel frame, such as in the stud flanges size and shape, may have a significant influence on LSF elements [19].

As mentioned before, another important heat transfer mechanism is convection. Therefore, another strategy to retain heat inside buildings, particularly in cold climates, is ensuring good airtightness of the building envelope [20], as well as inside the building elements, by using a continuous air barrier. Notice that besides thermal bridging, thermal bypass of cavities (convective heat flows through air leakage) could be also a weakness of LSF walls.

Moreover, regarding radiation thermal transfer, the use of thermal reflective insulation materials, such as a reflective low-emissivity foil or paint inside the air gaps of building elements (e.g., roofs and facade walls) is also an effective way to increase thermal resistance without increasing weight and wall thickness of these components, with the additional advantages of low cost and easy installation [21]. Jelle et al. [22] performed a very interesting state-of-the-art review about low emissivity materials for building applications for both opaque and transparent envelopes, such as windows, walls, roofs and floors.

Regarding the opaque building envelope, there are several strategies to take advantage of the radiation heat transfer mitigation, with the use of reflective foils [23] and paints [21] being the most established. In fact, the use of double pane elements (e.g., walls) with an air cavity, besides the extra thermal resistance provided by the air enclosure and a better humidity infiltration control, also allows to reduce the heat transfer by radiation whenever a low-emissivity surface is provided inside.

Bruno et al. [23] investigated the use of reflective thermal insulation in non-ventilated air gaps for refurbishment purposes, making use of exterior insulation panels. They measured an increase of the air gap thermal resistance up to seven times when commercial reflective panels were placed inside a non-ventilated air gap for the same wall sample thickness. Additionally, their numerical simulation results showed, for real-scale air cavities, that thermo-reflective panels can produce the same effect of at least 6 cm of traditional insulating materials (thermal conductivity of 0.030 W/(m·K)), with the advantage of avoiding the need of additional space. Moreover, they found an optimal air gap thickness of 4–5 cm when using low-emissivity reflective foils.

Fantucci and Serra [21] presented several solutions regarding the use of low-emissivity paints in opaque building envelopes, with the purpose of improving their thermal performance by reducing heat losses, given the smaller radiation heat exchange; this research work was mainly experimental. In this work, several case studies were presented, such as: (i) hollow bricks coated with low-emissivity paint; (ii) low-emissivity paint coating below roof tiles, and; (iii) low-emissivity paint on the walls behind radiators. Regarding these case studies, their experimental results allowed the following conclusions: (i) the equivalent thermal conductivity of the hollow bricks was reduced by 18%; (ii) the summer heat loads across a roof component were reduced by 19%, and; (iii) the heat loss from the wall behind radiators was reduced by 25%.

As recently stated by Bruno et al. [23], there is a lack of investigations about heat transfer inside enclosures equipped with low-emissivity materials. This lack is even more noticeable in LSF double pane elements containing an air cavity with a reflective foil; no research work addressing this issue was found in the literature.

In this work, the performance of a thermal reflective insulation system, constituted by an aluminium foil placed inside an air cavity between a double pane LSF partition, is experimentally evaluated for different air gap thicknesses, ranging from 0 mm up to 50 mm, with a step increment of 10 mm. First, the description of materials and methods are performed, including the LSF walls description and materials characterization. In this section, the experimental lab tests are also presented, together with the experimental setup, set points, test procedures and verifications. Moreover, the description of the numerical simulations used to verify and compare with the measurement results are carried out, namely the geometry, domain discretization, boundary conditions and unventilated airspaces models. Next, the obtained results are presented and discussed, and related to both lab measurements (thermal resistances and infrared thermography) and numerical simulations. Finally, the key conclusions of the present work are listed.

## 2. Materials and Methods

The aim of this section is to describe the tested LSF double pane walls and characterize the used materials. Moreover, the experimental setup is explained, and the test procedures described. Finally, a numerical simulation verification is performed to ensure the accuracy and reliability of the achieved experimental results.

### 2.1. Walls Description

Figure 1 displays a horizontal cross-section of the double pane LSF wall internal partitions evaluated in the lab experiments. Each LSF wall pane has steel studs (C48 × 37 × 4 × 0.6 mm) with 400 mm spacing, filled between with mineral wool (MW) (48 mm thick). On the outer surface there are two gypsum plasterboard (GPB) panels with a total thickness of 25 mm. The air cavity thickness is variable and changes from 50 mm down to 0 mm, with 10 mm step. Moreover, another variable parameter was the use of a reflective aluminium foil (emissivity of 0.05) on the outer surface of the air cavity.

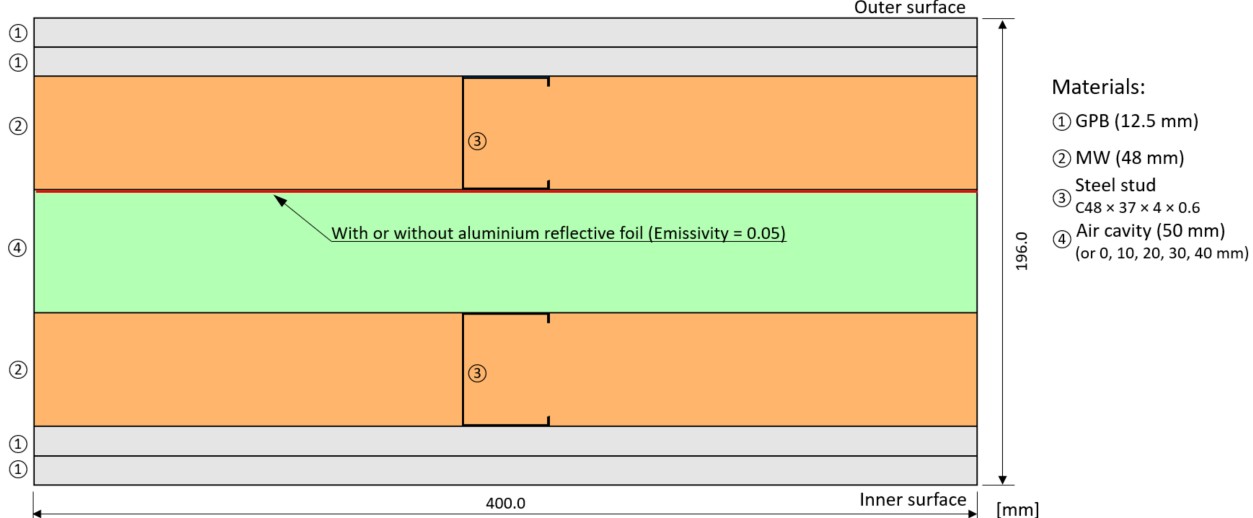

**Figure 1.** Ideal geometry and materials for a double pane LSF wall with different air cavity thicknesses, with and without a reflective aluminium foil.

### 2.2. Materials Characterization

Table 1 shows the thickness and the materials' thermal conductivities considered in the double pane LSF wall internal partitions tested in this research work. Notice that aluminium reflective foil thickness is very reduced (around 0.1 mm) and, therefore, its thermal conductive resistance was neglected. The emissivity of this aluminium reflective foil is 0.05 [23], the cold-formed galvanized steel is 0.23 [19], while for the remaining materials it is 0.90 [21].

**Table 1.** Double pane LSF wall materials, thickness (*d*) and thermal conductivities (λ).

| Material (Outer to Inner Layer) | *d* [mm] | λ [W/(m·K)] | Reference |
|---|---|---|---|
| GPB [1] (2 × 12.5 mm) | 25.0 | 0.175 | [24] |
| MW [2] | 48.0 | 0.035 | [25] |
| Steel stud (C48 × 37 × 4 × 0.6 mm) | - | 50.000 | [26] |
| Air cavity | 0, 10, 20, 30, 40, 50 | - | - |
| MW [2] | 48.0 | 0.035 | [25] |
| Steel stud (C48 × 37 × 4 × 0.6 mm) | - | 50.000 | [26] |
| GPB [1] (2 × 12.5 mm) | 25.0 | 0.175 | [24] |
| **Total Thickness** | **146.0–196.0** | - | - |

[1] GPB—Gypsum Plaster Board; [2] MW—Mineral Wool.

### 2.3. Experimental Lab Tests

#### 2.3.1. Experimental Setup

To perform the lab measurements, a mini hot box apparatus was used, as illustrated in Figure 2a. The heating of the hot box and the cooling of the cold box were carried out using an electrical resistance and a refrigerator, respectively. The double pane LSF wall test sample (Figure 2b) was placed between these two chambers. The heat loss that occurred through the lateral surfaces of the test sample was minimized by covering the perimeter with polyurethane foam insulation (80 mm thick), as showed in Figure 2a. The two LSF wall panes were separated by an EPS frame as illustrated in Figures 2a and 3b. Notice that the perimeter thickness of this EPS frame (Figure 3b) is equal to the thermal insulation thickness of the cold/hot box walls and slab envelopes.

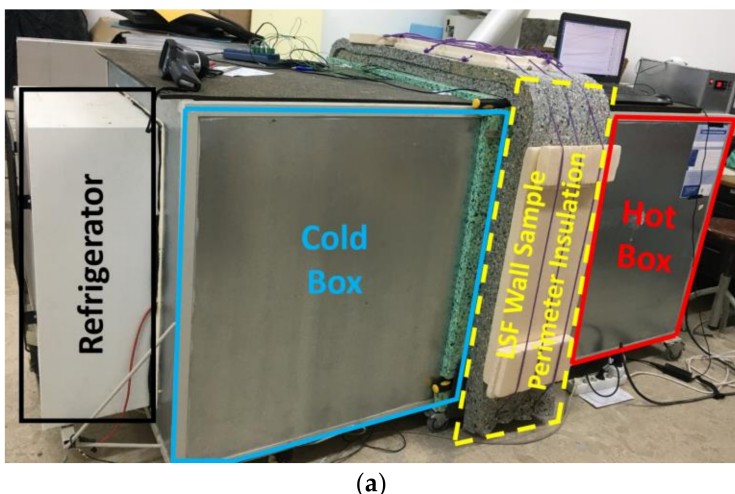

(a)

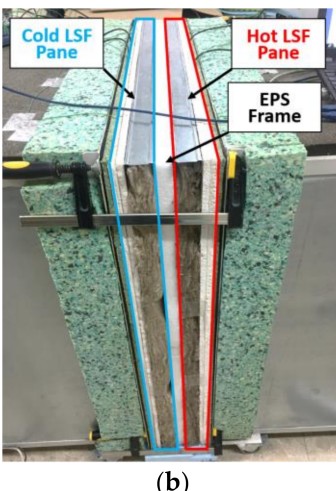

(b)

**Figure 2.** Mini hot box apparatus used in the lab measurements. (**a**) Lateral view with sample perimeter insulation; (**b**) LSF wall test sample.

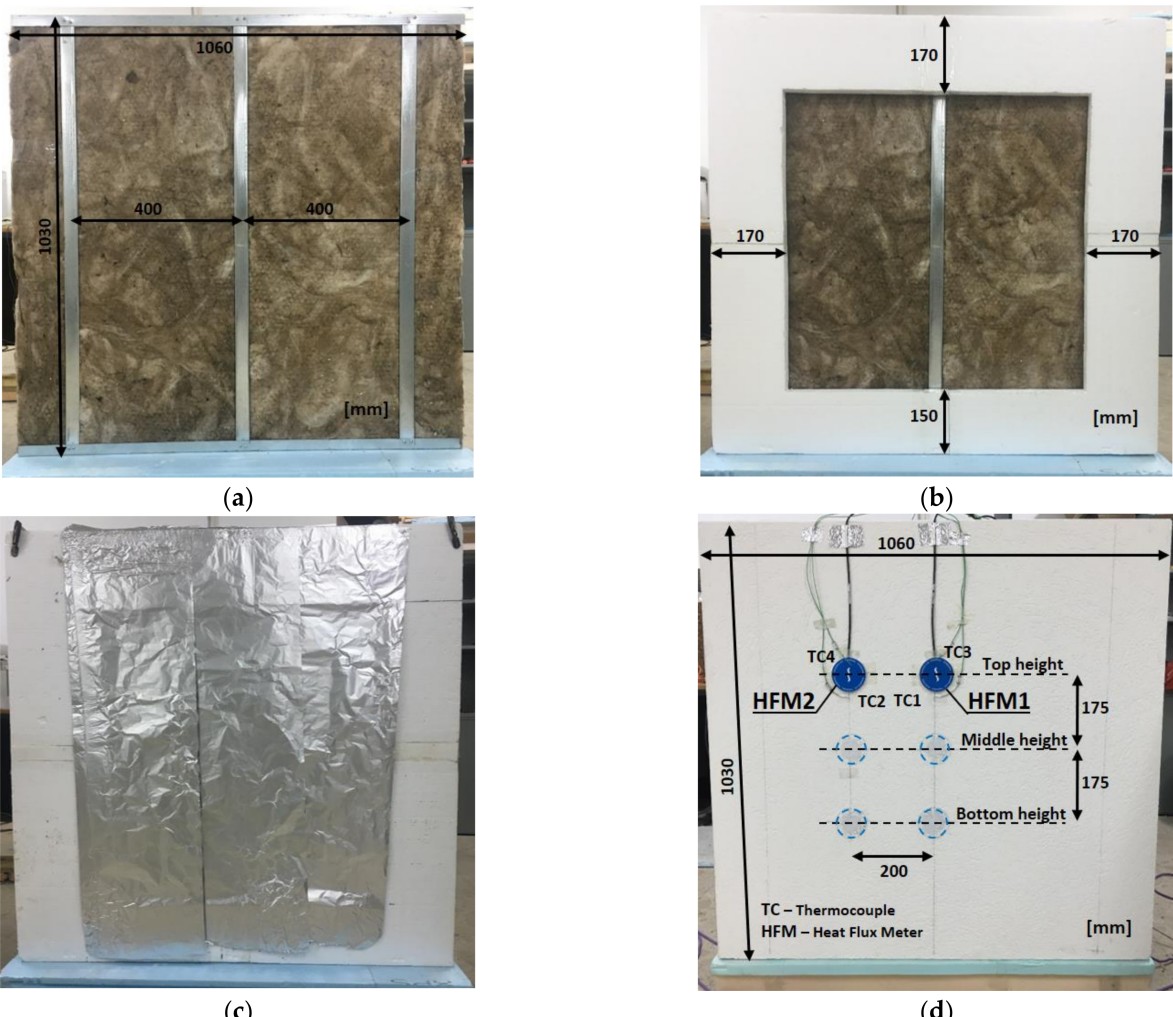

**Figure 3.** Double pane LSF wall test sample. (**a**) Hot LSF wall pane; (**b**) EPS perimeter frame; (**c**) Aluminium reflective foil; (**d**) Sensors (cold surface).

Figure 3a displays a frontal view of the LSF wall pane (hot side), where the vertical steel studs spacing is visible (400 mm), which is filled with mineral wool. Notice that among the three vertical steel studs, only the central one is visible (Figure 3b) and exposed to the temperature gradient established by the hot and cold boxes. The aluminium reflective foil was placed on the exterior (cold) side of the air cavity, as illustrated in Figure 3c. To promote internal air circulation and minimize the probability of air temperature stratification inside the cold and hot boxes, small interior fans were used. Moreover, two black radiation shields were used, one each side of the LSF wall test sample (10 cm apart).

Regarding the monitoring system, the heat flux through the test sample was measured using four heat flux meters (Hukseflux model HFP01, precision: $\pm 3\%$), of which two were placed on the hot wall surface and the another two were placed on the cold wall surface (Figure 3d). On the hot and cold wall surface, to measure the two distinct thermal behaviour zones that exist on the LSF wall sample, two locations for the application of the HFMs were considered: (1) in the central vertical steel stud zone (HFM1); and (2) midway through the insulation cavity (HFM2). To measure temperatures, twelve type K (1/0.315) PFA insulated thermocouples (TCs) were used, presenting class one precision certification. Furthermore, the calibration of these TCs was set in the temperature range [5 °C; 45 °C], with a 5 °C increment, by immerging the TCs in a thermostatic stirring water bath (Heto CB 208).

For each side of the experiment, six TCs were used to perform the measurements, and the following configuration has been defined: two measured the environment air temperature inside each box (TC5 and TC6), another two measured the air temperature between the radiation shield and the wall surface (TC3 and TC4), and the remaining two measured the wall surface temperatures (TC1 and TC2), as illustrated in Figure 3d. For each side of the wall test sample (hot and cold), one PICO TC-08 data logger (precision: ±0.5 °C) was used to record the temperature and heat flux data measured during the tests. The two data loggers were connected to a laptop and the data were managed using the PicoLog version 6.1.10 software. The main features of the measurement equipment used in the lab experiments are listed in Table 2.

**Table 2.** Features of the measurement equipment used in the lab experiments.

| Equipment | Brand | Model | Measurement Range | Precision |
|---|---|---|---|---|
| Heat Flux Meter | Hukseflux | HFP01 | −2 to +2 kW/m$^2$ | ±3% |
| Thermocouples | LabFacility | Type K * (1/0.315) | −75 to +260 °C | ±1.5 °C |
| Data-logger | PICO | TC-08 | −270 to +1820 °C | ±0.5 °C |

* Tolerance Class 1 certified.

Given the expansible behaviour of the MW batt insulation and since there was no confinement on the air cavity side, an increased thickness of this insulation layer was observed. As illustrated in Figure 4a, this increment in the MW insulation thickness was around 7.5 mm on each wall pane. Moreover, it was also observed that when there is no air cavity, the steel studs were not perfectly joined because the space is occupied by the heads of the connecting screws, as illustrated in Figure 4b. Therefore, it was assumed that the two contiguous steel studs were separated by 4 mm, when theoretically there is no air cavity between the LSF wall panes.

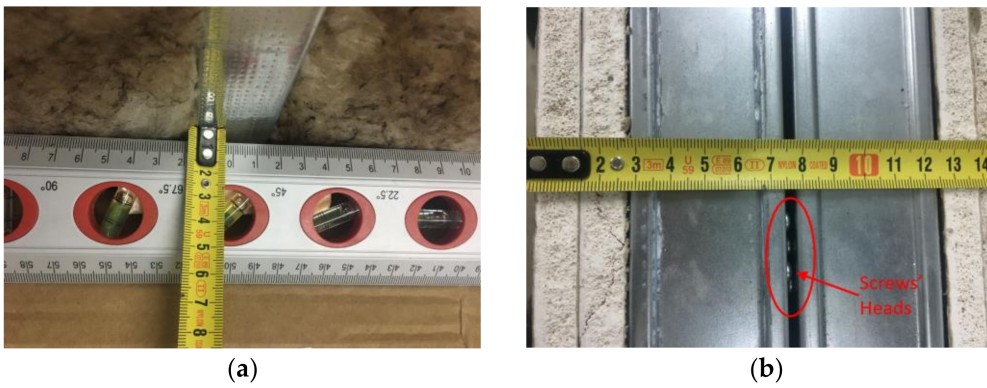

(**a**)  (**b**)

**Figure 4.** Geometry details of the measured double pane LSF wall test sample. (**a**) Expansible MW around steel stud; (**b**) Steel track distance.

2.3.2. Set-Points and Test Procedures

The measurement of the thermal performance of the LSF test samples was performed using the heat flow meter (HFM) method [27]. However, in order to increase the precision and reduce the test duration, instead of measuring on only one side (as prescribed by ISO 9869-1 [27]), the measurements were performed, simultaneously, at both wall surfaces (cold and hot), implementing the improvement suggested by Rasooli and Itard [28]. The hot and cold boxes were programmed to maintain set point temperatures of 40 °C and 5 °C, respectively, being the measurements performed in a quasi-steady-state heat transfer condition.

The convergence criteria prescribed in ASTM C1155–95 [29] were adopted for the "summation technique", i.e., assuming a maximum admissible convergence factor equal to 10%. Thus, only the estimated hourly *R*-values with an absolute difference, in relation to

the previous time obtained *R*-value, lower than 10% were considered in the measurements. Each measurement test was carried out for a minimum of 24 h.

To ensure the repeatability of the experimental measurements, for each wall three tests were performed corresponding to three high locations, as illustrated in Figure 3d, that is: (1) top, (2) middle, and (3) bottom. Furthermore, the average of these three tests were considered the measured overall conductive *R*-value of the LSF wall. From the heat fluxes and temperatures recorded for each test and applying the HFM method [27], two distinct conductive local *R*-values were obtained, corresponding to the two locations considered for the HFMs (Figure 3d): (1) a lower value in the central vertical steel stud zone ($R_{stud}$), and (2) a higher value in a midway through the insulation cavity ($R_{cav}$). To obtain the overall surface-to-surface -value of the wall ($R_{global}$), an area-weighted average of both measured conductive *R*-values were considered. Following the ASHRAE zone method [30], the steel stud influence area ($A_{stud}$), was defined considering a zone factor ($zf$) value of 2.0 [31]. More details about these measurements can be found in reference [14].

### 2.3.3. Test Procedures Verification

With the aim of ensuring the accuracy of the experimental apparatus (e.g., data-loggers and sensors) used in this work, a homogeneous XPS panel (Topox® Cuber SL), with a thickness of 60 mm and a thermal conductivity of 0.034 W/(m·K), were tested under the same conditions. The thermal conductive resistance measured (1.784 m²·K/W) indicates that the thermal conductivity value calculated experimentally is equal to the provided by the XPS manufacturer (0.034 W/(m·K)), confirming the good working conditions of the sensors and data acquisition system.

### 2.4. Numerical Simulation Verification

The finite element method (FEM) software THERM® (version 7.6.1) [32] was used to perform another verification of the measured *R*-value results, as explained next. This software is a well-known state-of-the-art freeware computer program for a two-dimensional heat-transfer analysis. It was developed in the United States of America (USA) by the Department of Energy (DoE), through the Lawrence Berkeley National Laboratory (LBNL).

### 2.4.1. Geometry and Domain Discretization

Since it is a bi-dimensional FEM numerical simulation due to the existence of only vertical steel studs (Figure 3a), only a 2D representative part of the double-pane LSF wall cross-section (400 mm width) was modelled, as previously illustrated in Figure 1. Nevertheless, in a previous research work [19] a similar 2D THERM model were compared with a 3D ANSYS model and, as expected, the thermal resistance difference between both models was very small, i.e., only 0.002 m² K/W. Therefore, in this work it was decided not to present a similar redundant comparison.

As previously mentioned in Section 2.3.1 and illustrated in Figure 4a, the MW insulation naturally expands. Since there was no confinement on the air cavity side, the ideal geometry model (Figure 1) was discarded. A new and more realistic model was adopted, as illustrated in Figure 5a, for a nominal 50 mm air cavity thickness, assuming a MW expansion of 7.5 mm for each insulation layer. Similarly, as previously illustrated in Figure 4b, the real steel studs are not perfectly united even for a nominal 0 mm air cavity thickness. Therefore, the correspondent THERM model assumes a 4 mm separation between the vertical steel studs and the consequent MW expansion, as showed in Figure 5b. Notice that all other LSF wall models for nominal air cavities of 10, 20, 30 and 40 mm are derived from the previous two, making use of similar geometric rules (not illustrated here for sake of brevity).

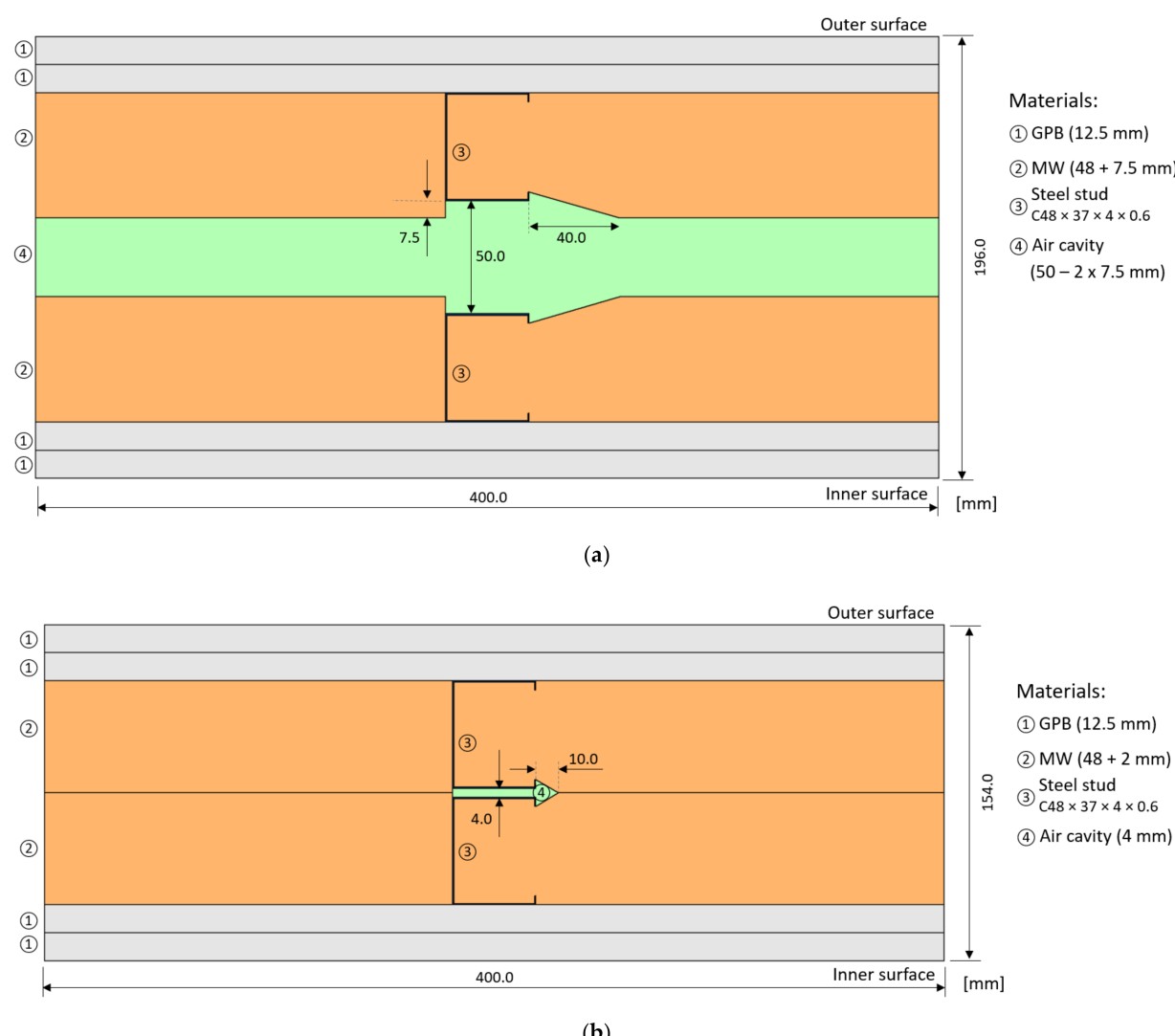

**Figure 5.** Modelled geometry and used materials for the double pane LSF walls. (**a**) Nominal 50 mm air cavity thicknesses; (**b**) Nominal 0 mm air cavity thicknesses.

Notice that the self-drilling screws used to assemble the steel frame (Figure 4b) and to fix the sheathing GPB to the studs were not modelled here. However, as demonstrated in previous research works [4], its relevance in the obtained thermal resistance value is neglectable.

The thermal properties of the materials used in these simulations were previously presented in Section 2.2. Additionally, the FEM mesh was refined to achieve a maximum error of 3% in these computations. The used quad tree mesh parameter was set to 6 and the maximum number of iterations was 100, the convergence tolerance was equal to $1 \times 10^{-6}$, and the mesh void tolerance was 1 mm$^2$. Using this mesh configuration, the maximum number of finite elements was 15,429.

### 2.4.2. Boundary Conditions

Regarding the boundary conditions, the air temperature was set to 40 °C and 5 °C, for the inner and outer environments, respectively. Notice that these values are equal to the set points defined for hot and cold boxes, as previously described in Section 2.3.2. Moreover, the surface thermal resistances were modelled using the average values measured for each LSF wall surface and for each test, considering the air and surface temperature differences and the surface heat fluxes. The measured surface thermal resistances vary within the interval [0.06; 0.13] m$^2$·K/W, thereby respecting the range defined by EN ISO 6946 [33] for

horizontal heat flow, i.e., between 0.04 m$^2$·K/W for external surface resistance ($R_{se}$) and 0.13 m$^2$·K/W for internal surface resistance ($R_{si}$).

In this work, since only the surface-to-surface (or conductive) *R*-values are considered to evaluate the thermal performance of the test samples, the surface thermal resistances are not included. However, they should be defined in the numerical simulations performed by THERM software, considering a film coefficient ($1/R_s$) being later deducted.

### 2.4.3. Unventilated Airspaces Models

The airspaces between the two LSF wall panes were modelled as unventilated making use of two different approaches available in the THERM software: CEN simplified and NFRC 100. Both methods make use of a solid-equivalent effective thermal conductivity of the airspace and incorporates the convective and radiative heat transfer effects by taking into account the geometry, heat flow direction, surface emissivity and temperature of the surrounding surfaces of the cavity area. The CEN simplified method is based on the calculation procedures defined in the standard EN ISO 6946 [33] for unventilated airspaces, while the NFRC 100 method is based on the simplified radiation model defined in the standard ISO 15099 [34].

## 3. Results and Discussion

### *3.1. Lab Measurements*

In this subsection, the lab-measured values are presented and discussed, including thermal resistances and the infrared thermography images.

### 3.1.1. Thermal Resistance Values

Figure 6 displays the measured conductive *R*-values of double pane LSF walls with different air cavity thicknesses (0–50 mm with an increment of 10 mm), with and without an aluminium reflective foil. Looking first to the -values without a reflective foil (black line), it is visible that the increment in the air cavity thickness is useful up to 30 mm, with the measured thermal resistance being nearly constant from there, i.e., for 40 and 50 mm. Adding a reflective aluminium foil (red line), there is a significant increase in the achieved *R*-value, particularly for higher thicknesses of the air cavity.

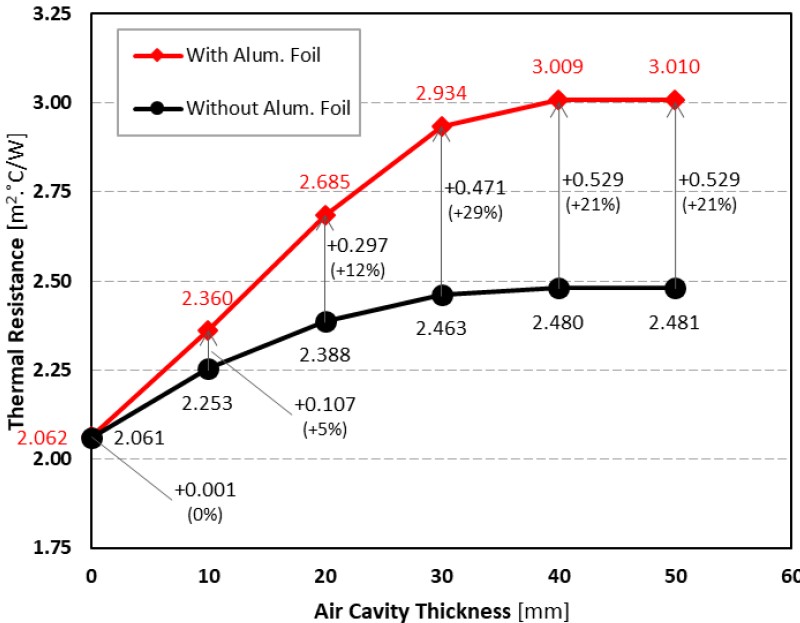

**Figure 6.** Measured conductive thermal resistances of a double pane LSF wall, with different air cavity thicknesses, with and without a reflective aluminium foil.

As also displayed in Figure 6, this thermal resistance increase ranges from only +0.001 m²·°C/W for a null air cavity thickness, up to + 0.529 m²·°C/W (+21%) for 50 mm air cavity thickness. When there is a reflective aluminium foil (red plot), the increment of the measured *R*-values with the increase in the air cavity thickness is almost linear until 30 mm, where it is nearly constant after 40 mm thick. Thus, it can be concluded that 30 mm is the recommended air cavity thickness when there is no reflective foil, while when using an aluminium reflective foil, the suggested thickness is increased to 40 mm. These results are in line with the conclusions achieved by Bruno et al. [23], which reported a maximum air gap thermal resistance, when using a reflective foil, for an air cavity thickness of 4 cm.

Notice that the maximum increased *R*-value measured (+0.529 m²·°C/W), when using an aluminium reflective foil, is equivalent to an 18.5 mm mineral wool layer (thermal conductivity equal to 0.035 W/(m·K)), but without the need of an increased wall thickness, whenever an air gap exists before the application of the foil. Additionally, the reflective aluminium foil thermal resistance increment (+0.529 m²·°C/W) is higher than the -value increase due to a 50 mm air gap when there is no reflective foil (+0.420 m²·°C/W), highlighting the importance of the radiative heat transfer. Furthermore, filling the cavity with mineral wool appears to have a comparable effect to the aluminium foil for small cavities. However, the existence of an air gap has several advantages, mainly related with humidity control in case of water infiltration.

### 3.1.2. InfraRed Thermography

Figure 7 shows the infrared (IR) images captured on the cold surface of the double pane LSF walls with and without a reflective aluminium foil, when using two extreme air cavity thicknesses: 0 and 50 mm. In these IR images, the central vertical steel stud is quite well visible, and it is more evident in the two LSF walls with null air cavity thicknesses. This is due to an increased localised heat transfer near the vertical steel stud, originating an augmented temperature in the cold surface of the wall.

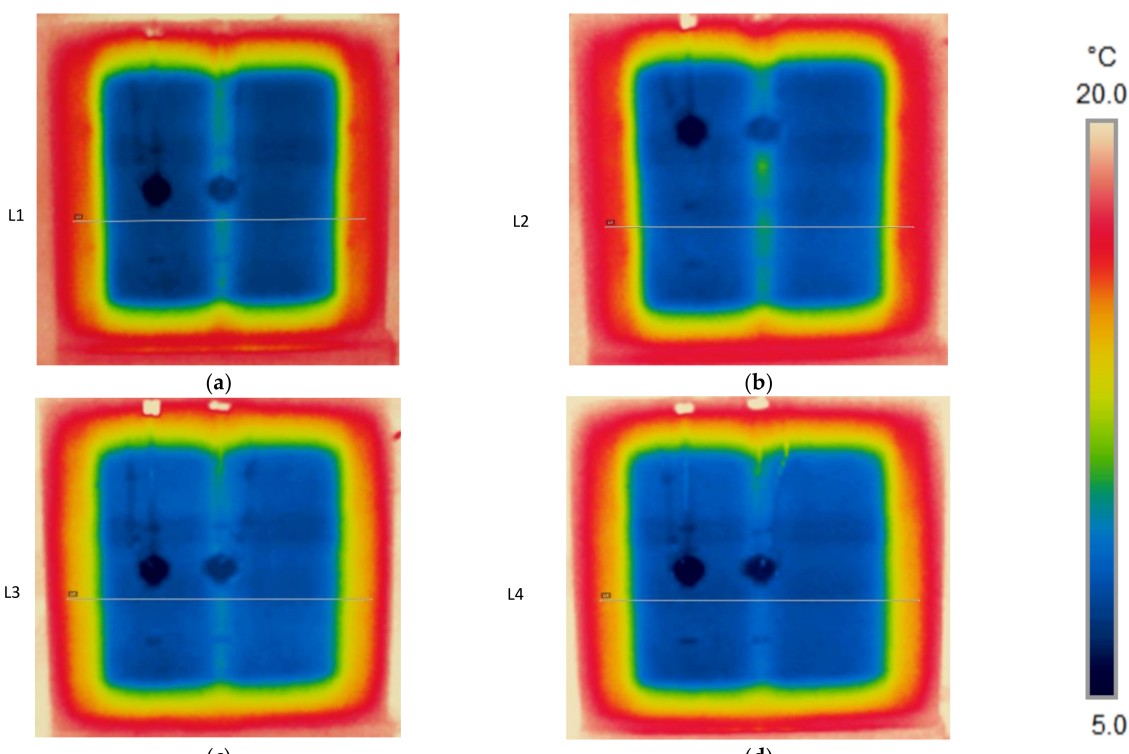

**Figure 7.** Infrared images of the double pane LSF walls, with and without reflective aluminium foil, for two air cavity's thicknesses (0 and 50 mm), on the cold surface. (**a**) 0 mm without reflective foil; (**b**) 0 mm with reflective foil; (**c**) 50 mm without reflective foil; (**d**) 50 mm with reflective foil.

To better visualize this steel stud thermal bridge effect, Figure 8 illustrates the surface temperatures recorded along horizontal lines in the IR images plotted in Figure 7. The highest peak temperatures were recorded near the steel stud for a null air cavity thickness, 10.6 °C and 10.0 °C, with and without an aluminium reflective foil, respectively. Another interesting feature visible in Figures 7 and 8 is that impact or influence of the thermal bridge originated by the vertical steel studs is well beyond the measurement circular area of the HFM1, which is only $8 \times 10^{-4}$ m$^2$ (diameter equal to 32 mm).

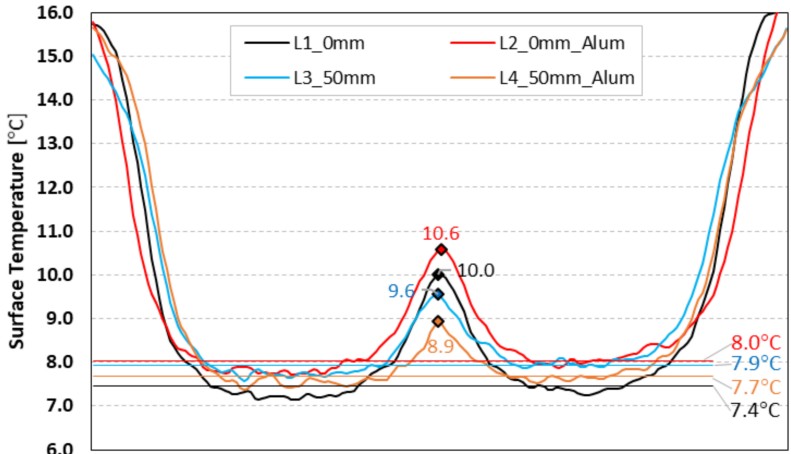

**Figure 8.** Horizontal temperature lines from infrared images of the double pane LSF walls, with and without reflective aluminium foil, for two air cavity's thicknesses (0 and 50 mm), on the cold surface.

Figure 9 shows the relative steel stud surface temperature increase along the horizontal lines on the cold surface of these four double pane LSF walls. Now, the higher surface temperature increase for the null air cavity thickness LSF walls is even better visualized due to the thermal bridge effect. In this case, the surface temperature increase near the vertical steel studs is +2.6 °C and +2.5 °C, with and without an aluminium reflective foil, respectively. As expected, for a 50 mm air cavity thickness the steel stud thermal bridge effect is quite smaller, exhibiting a surface temperature increase of only +1.6 °C and 1.3 °C, with and without aluminium reflective foil, respectively.

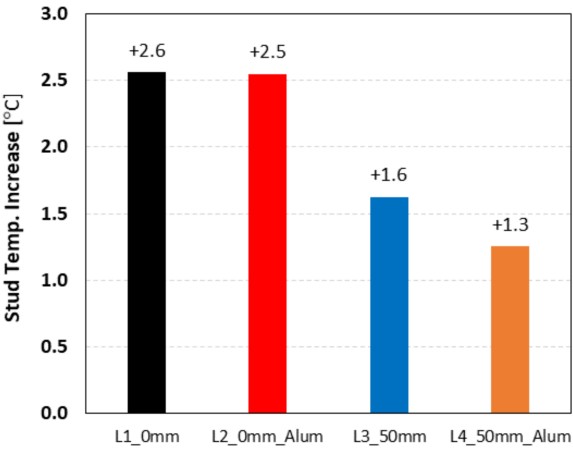

**Figure 9.** Steel stud surface temperature increase along horizontal lines on the cold surface of the double pane LSF walls, with and without reflective aluminium foil, for two air cavity's thicknesses (0 and 50 mm).

### 3.2. Numerical Simulations

As previously mentioned in Section 2.4, the experimental lab measurements were also compared with finite element numerical simulations predicted values making use of two

different approaches to model the air cavities: CEN simplified and NFRC 100. Table 3 displays the predicted *R*-values for both methodologies, with and without an aluminium reflective foil, for the evaluated air cavity thicknesses (0–50 mm), as well as the differences to the measured values. To better visualise this info, Figure 10 graphically illustrates the measured and the predicted CEN and NFRC *R*-values. The first remark is that both models can reproduce with quite good accuracy a similar trend to the measured *R*-values, by predicting for some air cavities higher and for others smaller *R*-values.

**Table 3.** Predicted conductive thermal resistances (*R*-values) of a double pane LSF wall, with different air cavity thicknesses, with and without a reflective aluminium foil, and the thermal resistance difference to the measured values.

| | Air Cavity | Without Aluminium Foil | | | With Aluminium Foil | | |
|---|---|---|---|---|---|---|---|
| | Thickness | *R*-Value | Difference | | *R*-Value | Difference | |
| | [mm] | [m$^2 \cdot °$C/W] | [m$^2 \cdot °$C/W] | [%] | [m$^2 \cdot °$C/W] | [m$^2 \cdot °$C/W] | [%] |
| **CEN Simplified** | 0 | 2.109 | +0.048 | +2% | 2.132 | +0.070 | +3% |
| | 10 | 2.242 | −0.011 | 0% | 2.424 | +0.064 | +3% |
| | 20 | 2.391 | +0.003 | 0% | 2.840 | +0.155 | +5% |
| | 30 | 2.391 | −0.072 | −3% | 2.993 | +0.059 | +2% |
| | 40 | 2.367 | −0.113 | −5% | 2.973 | −0.036 | −1% |
| | 50 | 2.354 | **−0.127** | **−5%** | 2.960 | −0.050 | −2% |
| **NFRC 100** | 0 | 2.105 | +0.044 | +2% | 2.125 | +0.063 | −3% |
| | 10 | 2.229 | −0.024 | −1% | 2.375 | +0.015 | −1% |
| | 20 | 2.388 | 0.000 | 0% | 2.826 | +0.141 | −5% |
| | 30 | 2.376 | −0.087 | −4% | 2.904 | −0.030 | −1% |
| | 40 | 2.350 | −0.130 | −6% | 2.881 | −0.128 | −4% |
| | 50 | 2.337 | **−0.144** | **−6%** | 2.863 | **−0.147** | **−5%** |

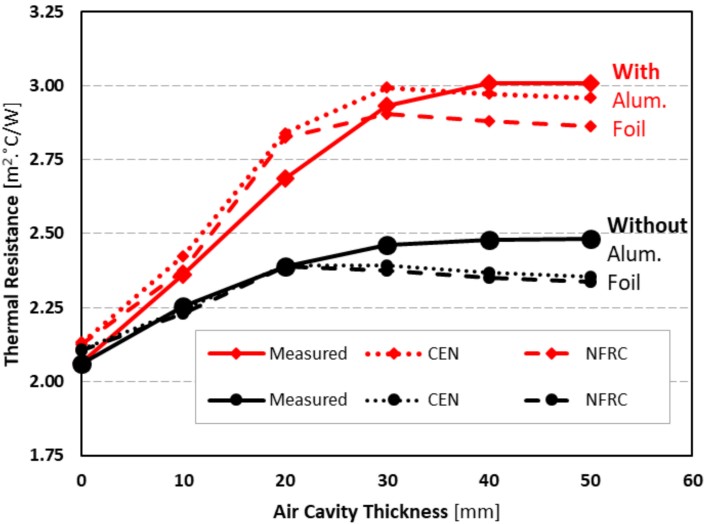

**Figure 10.** Measured and predicted conductive thermal resistances of a double pane LSF wall, with different air cavity thicknesses, with and without a reflective aluminium foil.

Without a reflective foil, both models predict very similar values, which are smaller than the measured ones for air cavities higher or equal to 30 mm. In this case, the major difference occurs for a 50 mm air gap, which is −0.127 m$^2 \cdot °$C/W (−5%) and 0.144 m$^2 \cdot °$C/W (−6%) for the CEN simplified and NFRC 100 models, respectively.

When there is an aluminium reflective foil, the differences between both models enlarge, particularly for higher thicknesses of the air cavity (30–50 mm). In this thickness range, the CEN simplified methodology provides results closer to the measured *R*-values. For instance, for 50 mm the predictions are −0.050 m$^2 \cdot °$C/W (−2%) and − 0.147 m$^2 \cdot °$C/W

(−5%) for CEN and NFRC, respectively. Moreover, for smaller air gaps (0–20 mm), these models exhibit a trend to slightly overestimate the *R*-values, ranging between +1% up to +5%.

## 4. Conclusions

In this work, the thermal performance of double-pane LSF walls with and without an aluminium reflective foil was assessed. This assessment was based in lab measurements and the recorded conductive *R*-values were compared with two different approaches to model the air cavities: "CEN simplified" [33] and "NFRC 100" [34], which were implemented in a 2D Finite Element software [32]. The thicknesses of these air gaps varied from 0 mm up to 50 mm, with an increment of 10 mm.

The major conclusions of this study could be summarized as follows:

- The use of a reflective foil is a very effective way to increase the thermal resistance of double pane LSF walls, without increasing the wall thickness and weight; the maximum improvement is around +0.529 $m^2 \cdot °C/W$ (+21%) in the achieved conductive *R*-value.
- It is not worthy to increase the air gap to values higher than 30 mm or 40 mm with and without a reflective foil, respectively.
- The *R*-value increase due to the reflective aluminium foil (+0.529 $m^2 \cdot °C/W$), is higher than the *R*-value raise due to a 50 mm air gap when there is no reflective foil (+0.420 $m^2 \cdot °C/W$).
- Besides the thermal resistance increase, the use of an aluminium reflective foil is also favourable to reduce the thermal bridge effect due to the steel studs' high thermal conductivity, whenever there are both a continuous air cavity and an aluminium foil.
- Both "CEN simplified" and "NFRC 100" models were able to predict with reasonable accuracy the thermal behaviour of the air cavities within the evaluated double pane LSF walls, ranging the obtained differences around ±5%.
- The major differences between both air cavity models arises for bigger air gaps (30–50 mm) when an aluminium reflective foil (0.05 emissivity) is used.

Notice that in this study the air cavity evaluated was continuous. Therefore, any decrease in thermal resistance due to wall ties, connectors or other bridging elements inside the cavity was not considered.

**Author Contributions:** Conceptualization, P.S.; methodology, P.S.; validation, P.S. and T.R.; formal analysis, P.S.; investigation, P.S. and T.R.; writing—original draft preparation P.S.; writing—review and editing, P.S. and T.R.; visualization, P.S.; supervision, P.S.; project administration, P.S.; funding acquisition, P.S. All authors have read and agreed to the published version of the manuscript.

**Funding:** This research was funded by FEDER funds through the Competitivity Factors Operational Programme—COMPETE and by national funds through FCT—Foundation for Science and Technology within the scope of the project POCI-01-0145-FEDER-032061.

Cofinanciado por: POCI-01-0145-FEDER-032061

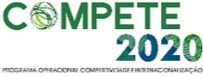 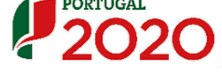 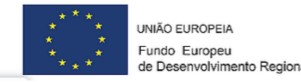 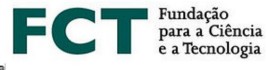

**Institutional Review Board Statement:** Not applicable.

**Informed Consent Statement:** Not applicable.

**Acknowledgments:** The authors also want to thank the support provided by the following companies: Pertecno, Gyptec Ibéria, Volcalis, Sotinco, Kronospan, Hulkseflux, Hilti and Metabo.

**Conflicts of Interest:** The authors declare no conflict of interest.

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
