# Peer review of "Thermal Performance of Double-Pane Lightweight Steel Framed Walls with and without a Reflective Foil"

_buildings, doi:10.3390/buildings11070301_

Round 1
Reviewer 1 Report
Abstract
… the most vulgar way. Please rephrase the text.
Introduction
Line 32: The most vulgar way… Please rephrase.
2.4. Numerical Simulation Verification
Give more details for the software THERM (number of elements used, BVP, conversion analysis ect.). Make additional comments on the comparisons you made with the experiments. Is three-dimensional analysis necessary for the problem? Make some comments. Give references.
Author Response
The authors would like to express their sincere gratitude to the reviewer for the wise constructive improvement suggestions, which have helped to improve the quality of the manuscript.
The authors took into consideration the suggestions of the reviewer and improved the paper as recommended.
Please find the point-by-point replies to the reviewer’s queries, as well as the revised manuscript with MS Word track-changes and a clean copy.
“Abstract
… the most vulgar way. Please rephrase the text.
Introduction
Line 32: The most vulgar way… Please rephrase.“
As suggested, both sentences were rephrased.
“2.4. Numerical Simulation Verification
Give more details for the software THERM (number of elements used, BVP, conversion analysis ect.). Make additional comments on the comparisons you made with the experiments. Is three-dimensional analysis necessary for the problem? Make some comments. Give references.”
The THERM software is a well-known state-of-the-art freeware computer program for a two-dimensional heat-transfer analysis, which was developed in the United States of America (USA) by the Department of Energy (DoE), through the Lawrence Barkley National Laboratory (LBNL).
Nevertheless, some more details were provided regarding this software, the models used (e.g., number of elements used and the neglected self-drilling screws), the 3D analysis and more related references were cited (e.g., references [4] and [19]).
Regarding the “three-dimensional analysis”, for these 2D heat transfer models there is no need to proceed with a 3D evaluation, as previously mentioned in lines 235-236: “Since it is a bi-dimensional FEM numerical simulation, only a 2D representative part of the double-pane LSF wall cross-section (400 mm width) was modelled”. Nevertheless, now it is explicitly mentioned the existence of only vertical steel studs and it is mentioned a previous work where this 2D-3D comparison was performed.
Concerning the comparison with experiments, the authors were not able to understand well if the reviewer was referring to Section 2.3.3 (Test Procedures Verification) or to Section 3.2 (Numerical Simulations), neither what kind of comments are being suggested. Sorry for this misunderstanding.
Reviewer 2 Report
The work presented in the article is interesting and relevant, especially the inclusion of experimental measurements of air cavities of different width and their comparison to currently accepted thermal models.
English language can be improved, even if the article is already well understandable.
The effect of low-emissivity materials and coatings is often misunderstood and this paper could contribute to a better understanding. Therefore, my biggest concern is the presentation of the conclusions.
The paper claims that an improvement in thermal resistance is achieved “without the need of an increased wall thickness” (e.g. line 314). This is only true also if a continuous, unventilated air gap would exist before the application of the foil is considered. Furthermore, looking at the figures in Table 2 (a very interesting outcome of the study), filling the cavity with mineral wool appears to have a comparable effect to the aluminium foil for small cavities (up to 40 mm) and would be more advantageous than the foil for larger cavities.
It is true (as well explained in the Introduction, e.g. line 40) that thermal bridges constitute a weak point of LSF walls, especially when steel studs bridge the insulation. But this also remains true for the case for reflective foils. The Conclusions state that “the use of an aluminium reflective foil is also favourable to reduce the thermal bridge effect” (line 391). This might be misleading: for achieving the impact claimed in the paper, the cavity must remain continuous. This is not clearly stated in the paper, and is also difficult to achieve in construction practice. Conclusions should explicitly recognise that the paper assumes a continuous air cavity, and that the decrease in thermal resistance due to any wall ties, connectors or other bridging elements in the cavity is not considered.
Just as a suggestion, for a clearer understanding to a general audience, the acronym LSF could be fully spelled (Lightweight Steel Frame?) in the title and in its first appearances in the abstract and the article text.
Line 6 and 32: Thermal insulation is defined as “the most vulgar way” to increase thermal resistance. The adjective “vulgar” has negative connotations and other alternatives (e.g. “simplest”, “most straightforward”) would be preferable.
Line 7: “The radiation heat transfer could be also very relevant”: this is true, but only in the presence of air cavities.
Line 21: In the abstract, it would be more useful to quote the improvement in thermal resistance (m²K/W or m²°C/W) instead of the +21%. The former can be extrapolated to other wall assemblies, while the percentage is very dependent on the specific assembly and insulation studied.
Line 61: “ensuring a good airtightness” is suggested instead of “ensuring a good air leakage”. It could be noted that, besides thermal bridging, thermal bypass of cavities (convective heat flows through air leakage) is also a weakness of LSF walls.
Line 62: It is suggested that these losses can be prevented “by ensuring that air gaps are not ventilated”. It would be clearer to say that these can be prevented by a continuous air barrier. If a continuous air barrier is in place, ventilated air cavities (e.g. behind cladding) could exist with no loss in the overall thermal resistance.
Line 217: The use of the zone factor for the influence area of the thermal bridge seems well justified and referenced. However, just out of interest, Figure 8 demonstrates that the impact of the thermal bridge extends well beyond the measurement area of the HFM1.
Figure 5: This approach is an interesting contribution by the authors. The use of more ‘realistic’ models to capture ‘as built’ conditions instead of simple assumptions is interesting and relevant.
Figure 7, 8, 9 and captions: The use of mm instead of cm would be preferable.
Line 333 and Figure 8: This is an important comment. Showing a percentage increase in °C temperature is conceptually incorrect, since 0 °C is an arbitrary threshold. Absolute values are already interesting, so Figure 9b could be deleted together with all references to percentage temperature increase in the text of the paper.
Author Response
The authors would like to express their sincere gratitude to the reviewer for the careful and thorough reading of the paper and for the thoughtful and useful comments, and wise constructive improvement suggestions, which have helped to improve the quality of the manuscript.
The authors took into consideration the suggestions of the reviewer, and a big effort was carried out to improve the paper as recommended.
Please find the point-by-point replies to the reviewer’s queries, as well as the revised manuscript with MS Word track-changes and a clean copy.
“The work presented in the article is interesting and relevant, especially the inclusion of experimental measurements of air cavities of different width and their comparison to currently accepted thermal models.”
Thank you very much.
“English language can be improved, even if the article is already well understandable.
The effect of low-emissivity materials and coatings is often misunderstood and this paper could contribute to a better understanding. Therefore, my biggest concern is the presentation of the conclusions.
The paper claims that an improvement in thermal resistance is achieved “without the need of an increased wall thickness” (e.g. line 314). This is only true also if a continuous, unventilated air gap would exist before the application of the foil is considered. Furthermore, looking at the figures in Table 2 (a very interesting outcome of the study), filling the cavity with mineral wool appears to have a comparable effect to the aluminium foil for small cavities (up to 40 mm) and would be more advantageous than the foil for larger cavities.”
Yes, we agree with the reviewer: “This is only true also if a continuous, unventilated air gap would exist before the application of the foil is considered” and this information was added to the manuscript.
We also agree with the reviewer, regarding: “filling the cavity with mineral wool appears to have a comparable effect to the aluminium foil for small cavities”. However, the existence of an air-gap has several advantages, mainly related with humidity control in case of water infiltration. These informations were also added to the manuscript.
“It is true (as well explained in the Introduction, e.g. line 40) that thermal bridges constitute a weak point of LSF walls, especially when steel studs bridge the insulation. But this also remains true for the case for reflective foils. The Conclusions state that “the use of an aluminium reflective foil is also favourable to reduce the thermal bridge effect” (line 391). This might be misleading: for achieving the impact claimed in the paper, the cavity must remain continuous. This is not clearly stated in the paper, and is also difficult to achieve in construction practice. Conclusions should explicitly recognise that the paper assumes a continuous air cavity, and that the decrease in thermal resistance due to any wall ties, connectors or other bridging elements in the cavity is not considered.”
Yes, we agree with the reviewer, i.e., the sentence in line 391 is only true when the cavity remains continuous. This is now explicitly mentioned in the Conclusions.
“Just as a suggestion, for a clearer understanding to a general audience, the acronym LSF could be fully spelled (Lightweight Steel Frame?) in the title and in its first appearances in the abstract and the article text.”
As suggested, the acronym “LSF” is now written in full the first time it is mentioned in the abstract and article text, as well as in the title of this draft paper.
“Line 6 and 32: Thermal insulation is defined as “the most vulgar way” to increase thermal resistance. The adjective “vulgar” has negative connotations and other alternatives (e.g. “simplest”, “most straightforward”) would be preferable.”
It was not our objective to give a negative connotation to the word “vulgar”. Nevertheless, as suggested the expression “most vulgar way” was replaced by: “simplest and most straightforward way”.
“Line 7: “The radiation heat transfer could be also very relevant”: this is true, but only in the presence of air cavities.”
Yes, we agree with the reviewer and this info is now explicitly mentioned in the article text.
“Line 21: In the abstract, it would be more useful to quote the improvement in thermal resistance (m²K/W or m²°C/W) instead of the +21%. The former can be extrapolated to other wall assemblies, while the percentage is very dependent on the specific assembly and insulation studied.”
As suggested, besides the percentage value, it was added an absolute value for the achieved thermal resistance increase.
“Line 61: “ensuring a good airtightness” is suggested instead of “ensuring a good air leakage”. It could be noted that, besides thermal bridging, thermal bypass of cavities (convective heat flows through air leakage) is also a weakness of LSF walls.”
Thanks for the typo error correction. The above-mentioned additional drawback of LSF walls is now mentioned in the manuscript.
“Line 62: It is suggested that these losses can be prevented “by ensuring that air gaps are not ventilated”. It would be clearer to say that these can be prevented by a continuous air barrier. If a continuous air barrier is in place, ventilated air cavities (e.g. behind cladding) could exist with no loss in the overall thermal resistance.”
As suggested, the sentence was reformulated.
“Line 217: The use of the zone factor for the influence area of the thermal bridge seems well justified and referenced. However, just out of interest, Figure 8 demonstrates that the impact of the thermal bridge extends well beyond the measurement area of the HFM1.”
Yes, it is true and this is now also mentioned in the draft article.
“Figure 5: This approach is an interesting contribution by the authors. The use of more ‘realistic’ models to capture ‘as built’ conditions instead of simple assumptions is interesting and relevant.”
Thank you very much.
“Figure 7, 8, 9 and captions: The use of mm instead of cm would be preferable.
Line 333 and Figure 8: This is an important comment. Showing a percentage increase in °C temperature is conceptually incorrect, since 0 °C is an arbitrary threshold. Absolute values are already interesting, so Figure 9b could be deleted together with all references to percentage temperature increase in the text of the paper.”
As suggested, the “cm” units were replaced by “mm”. Moreover, Figure 9b was deleted, as well as all the percentage temperature increase values mentioned along the text.
Reviewer 3 Report
In this study titled “Thermal performance of double-pane LSF walls with and without a reflective foil”, the authors basically evaluated the R-value (thermal resistivity) of a cavity wall with and without Al foil. The authors considered air gaps ranging from 0mm to 50mm. For the most part, the study is interesting and well structured, but certain sections need improvements. Also, the context requires clear clarifications. Please find some comments below to improve the quality and reliability of the study:
1. What is the main difference between this study and studies done by Bruno et al. (Reflective thermal insulation in non-ventilated airgaps: experimental and theoretical evaluations on the global heat transfer coefficient, Energy Build., vol. 236, p. 110769, 2021)? Bruno et al. is also cited many times in different sections as Ref. 22 in this study. Bruno et al. also conducted an experimental campaign in a climatic chamber and investigated three different samples of commercial reflective panels inside a non-ventilated airgap. What is more, this current study adopted similar procedures and parameters as used by Bruno et al. [22]. Even one key result that R-values are constant when the air gap is increased beyond 40mm, is similar to results in Bruno et al. [22]. Thus, the authors even stated that their results were in line with the results in Bruno et al (line 303). Apart from differences in the tested wall configurations in the two studies, please what is the substantial and clear difference between your study and Bruno et al.’s study? What makes this study distinct from Ref. [22]?
2. In the Abstract, “The use 5 of thermal insulation is the most vulgar way to promote thermal resistance of building…”. The use of ‘vulgar’ here is not appropriate (also used in the introduction -line 32). Please check and correct. Please explains LSF in full, the first time it is used in the abstract.
3. Concerning measurement equipment, it would be better to summarize the model types, accuracies/precision and uncertainties, measurement ranges, and other relevant specifications in a Table. This makes for easy comprehension. Currently, only the precision of heat flux meters (line 161) and data logger (line 180) is mentioned in different paragraphs of the text.
4. What is the difference between Table 2 and Figure 6? They represent the same information. Please maintain one and delete the other. The trends for increasing thermal resistance with and without Al foil are clearly visible already in Table 6.
5. Often, numerical models and validated by experimental results. And then based on the validated numerical models, further analysis is conducted. In this study, the numerical results via THERM software showed a very good correlation with experimental results. But what is the main reason for carrying out the numerical modeling? Because no further analysis is done afterward. Is it to show compliance to NRFC 100 and CEN simplified in THERM software only?
6. Significantly concerning sentence style or structure there are too many long sentences that run for 3-4 lines. Authors have connected these long sentences using commas (,), colons (:), and semi-colons (;). This makes certain portions of the manuscript difficult to read. The manuscript will look much better and more understandable if these long sentences are broken down into shorter sentences. For example, instead of:
“The heating of the hot box and the cooling of the cold box were carried out using an electrical resistance and a refrigerator, respectively, and the double pane LSF 143 wall test-sample (Figure 2b) was placed between these two chambers” (lines 142-144).
It is better to say:
“The heating of the hot box and the cooling of the cold box were carried out using an electrical resistance and a refrigerator, respectively. And the double pane LSF wall test-sample (Figure 2b) was placed between these two chambers” - much simpler and better. There is absolutely no need to join these sentences. This is just one example. There are many sentences like this in the manuscript.
7. General grammar and coherency of the manuscript need improvement. For example:
“It was found a maximum improvement of the double pane LSF wall’s thermal resistance due to the reflective foil of around +21% (lines 13-14).
“Moreover, this thermal insulation material also promotes sound insulation, mainly when are used fibrous insulation materials inside the air gaps (lines 35-36).
“To promote internal air circulation and minimize the probability of air temperature stratification inside the cold and hot boxes, it was used small interior fans (lines 157-158).
Please try to proofread the manuscript thoroughly.
Author Response
The authors would like to express their sincere gratitude to the reviewer for the careful and thorough reading of the paper and for the thoughtful and useful comments, and wise constructive improvement suggestions, which have helped to improve the quality of the manuscript.
The authors took into consideration the suggestions of the reviewer, and a big effort was carried out to improve the paper as recommended.
Please find the point-by-point replies to the reviewer’s queries, as well as the revised manuscript with MS Word track-changes and a clean copy.
“In this study titled “Thermal performance of double-pane LSF walls with and without a reflective foil”, the authors basically evaluated the R-value (thermal resistivity) of a cavity wall with and without Al foil. The authors considered air gaps ranging from 0mm to 50mm. For the most part, the study is interesting and well structured, but certain sections need improvements. Also, the context requires clear clarifications. Please find some comments below to improve the quality and reliability of the study:
1. What is the main difference between this study and studies done by Bruno et al. (Reflective thermal insulation in non-ventilated airgaps: experimental and theoretical evaluations on the global heat transfer coefficient, Energy Build., vol. 236, p. 110769, 2021)? Bruno et al. is also cited many times in different sections as Ref. 22 in this study. Bruno et al. also conducted an experimental campaign in a climatic chamber and investigated three different samples of commercial reflective panels inside a non-ventilated airgap. What is more, this current study adopted similar procedures and parameters as used by Bruno et al. [22]. Even one key result that R-values are constant when the air gap is increased beyond 40mm, is similar to results in Bruno et al. [22]. Thus, the authors even stated that their results were in line with the results in Bruno et al (line 303). Apart from differences in the tested wall configurations in the two studies, please what is the substantial and clear difference between your study and Bruno et al.’s study? What makes this study distinct from Ref. [22]?”
The only similarity between this study and the one published by Bruno et al. (former Ref. [22], now Ref. [23] is the research topic, i.e., reflective thermal insulation. Thus, the main differences between these studies are listed next:
a) Our study is mainly experimental, while in Ref. [22] is mainly a parametric study.
b) Here we measured a double pane LSF walls with different air-gap thicknesses (0-50 mm), while in Ref. [22] it is assessed a double wooden panel with a constant air-gap thickness (100 mm).
c) Our wall test-sample has an area of 1030 x 1060 mm2 with a total thickness ranging from 146 mm up to 196 mm, while in Ref. [22] the testing sample is 900 x 400 mm2 with a constant global thickness of 116 mm.
d) Our temperature difference between the hot and cold boxes was 35 °C, while in Ref. [22] they used 3 different test conditions with temperature differences equal to 10, 15 and 20 °C.
e) We placed our sensors (heat flux meters and thermocouples) at 3 different heights/locations in the wall test-specimen (top, middle and bottom), while they only measured in the middle/centre.
f) We placed heat flux meters at both sides of the wall test-sample (hot and cold), while they only placed on the hot side.
g) Our numerical simulations were performed making use of THERM software, while they used the COMSOL software.
h) The thermal resistance provided by the air cavities was predicted by two analytical methods (CEN Simplified and NFRC 100) incorporated in the THERM software, while they make use of a Computational Fluid Dynamics (CFD) algorithm from COMSOL software.
These are some of the main differences between these two studies. Thus, can be easily concluded that these two research works are completely distinct, without any overlapping.
“2. In the Abstract, “The use 5 of thermal insulation is the most vulgar way to promote thermal resistance of building…”. The use of ‘vulgar’ here is not appropriate (also used in the introduction -line 32). Please check and correct. Please explains LSF in full, the first time it is used in the abstract.”
As suggested, the expression “most vulgar way” was replaced, being now: “simplest and most straightforward way”. Moreover, the abbreviation “LSF” is now written in full the first time it is mentioned in the abstract, as well as in the title of this draft paper.
“3. Concerning measurement equipment, it would be better to summarize the model types, accuracies/precision and uncertainties, measurement ranges, and other relevant specifications in a Table. This makes for easy comprehension. Currently, only the precision of heat flux meters (line 161) and data logger (line 180) is mentioned in different paragraphs of the text.”
As suggested, a new table was added to the manuscript with the main features of the measurement equipment.
“4. What is the difference between Table 2 and Figure 6? They represent the same information. Please maintain one and delete the other. The trends for increasing thermal resistance with and without Al foil are clearly visible already in Table 6.”
Besides the graphical representation in Figure 6, the only differences are in the last 2 columns of Table 2, i.e., the “R-value increase”.
Nevertheless, to avoid any unwanted repetition, the authors decided to remove Table 2 and add the missing information to Figure 6.
“5. Often, numerical models and validated by experimental results. And then based on the validated numerical models, further analysis is conducted. In this study, the numerical results via THERM software showed a very good correlation with experimental results. But what is the main reason for carrying out the numerical modeling? Because no further analysis is done afterward. Is it to show compliance to NRFC 100 and CEN simplified in THERM software only?”
Yes, we agree with the reviewer. Usually, the numerical models are validated with experimental results. Afterwards, this validated model is used to make, for instance, a parametric study.
However, in this study, which is mainly an experimental research work, the main focus was in the lab measurment results. Therefore, they are presented first.
Nevertheless, the authors decided that would be interesting to compare the available measured values with the predicted simulated results, provided by two different methods. This way, it was possible not only to verify the accuracy of these analytical methods, as well as to ensure the reliability of the measurements.
“6. Significantly concerning sentence style or structure there are too many long sentences that run for 3-4 lines. Authors have connected these long sentences using commas (,), colons (:), and semi-colons (;). This makes certain portions of the manuscript difficult to read. The manuscript will look much better and more understandable if these long sentences are broken down into shorter sentences. For example, instead of:
“The heating of the hot box and the cooling of the cold box were carried out using an electrical resistance and a refrigerator, respectively, and the double pane LSF 143 wall test-sample (Figure 2b) was placed between these two chambers” (lines 142-144).
It is better to say:
“The heating of the hot box and the cooling of the cold box were carried out using an electrical resistance and a refrigerator, respectively. And the double pane LSF wall test-sample (Figure 2b) was placed between these two chambers” - much simpler and better. There is absolutely no need to join these sentences. This is just one example. There are many sentences like this in the manuscript.”
Yes, we agree with the reviewer and deeply acknowledge the improvement suggestion to split long sentences into smaller ones. This procedure will allow to simplify and turning more easier their understanding. Therefore, the above-mentioned sentence was divided, and similar procedures were implemented along the text of this draft manuscript.
“7. General grammar and coherency of the manuscript need improvement. For example:
“It was found a maximum improvement of the double pane LSF wall’s thermal resistance due to the reflective foil of around +21% (lines 13-14).
“Moreover, this thermal insulation material also promotes sound insulation, mainly when are used fibrous insulation materials inside the air gaps (lines 35-36).
“To promote internal air circulation and minimize the probability of air temperature stratification inside the cold and hot boxes, it was used small interior fans (lines 157-158).
Please try to proofread the manuscript thoroughly.”
As suggested, the above-mentioned sentences were corrected, and the manuscript text revised and improved. The authors are not native English speakers. Therefore, we have some difficulties to ensure a good and irreprehensible written text.
Nevertheless, this is not an issue, since the MDPI editorial services provide a very deep and effective text revision when a draft manuscript is accepted for publication.
Therefore, we strongly believe that any remaining grammar incoherency will be, as usual, corrected posteriorly, in due time, and this manuscript will not be an exception.
Round 2
Reviewer 1 Report
-
Reviewer 3 Report
The authors have carried out the review work and answered all the reviewer questions and suggestions. Most importantly about what distinguishes this study from previous studies and particular about Figure 6 and Table 2 (in old manuscript version). However the English content and sentence structuring including grammar, were not improved to a satisfactory level. The authors claim it will be done in due course. The current contribution of the manuscript is okay to be published after a thorough proofreading and English editing review.